# Alternative *c-MYC* mRNA Transcripts as an Additional Tool for c-Myc2 and c-MycS Production in BL60 Tumors

**DOI:** 10.3390/biom12060836

**Published:** 2022-06-16

**Authors:** Dina Ibrahim, Léa Prévaud, Nathalie Faumont, Danielle Troutaud, Jean Feuillard, Mona Diab-Assaf, Ahmad Oulmouden

**Affiliations:** 1CNRS UMR-7276, INSERM U1262, CRIBL, Dupuytren Hospital University Center (CHU) of Limoges, University of Limoges, 87036 Limoges, France; dina.16.ibrahim@hotmail.com (D.I.); lea.prevaud@unilim.fr (L.P.); nathalie.faumont@unilim.fr (N.F.); danielle.troutaud@unilim.fr (D.T.); jean.feuillard@unilim.fr (J.F.); 2Department of Sciences, Lebanese University Fanar, Beirut 1500, Lebanon; mdiabassaf@ul.edu.lb

**Keywords:** lymphoma cell lines, lymph nodes, c-Myc protein, BL60 cell line

## Abstract

While studying c-Myc protein expression in several Burkitt lymphoma cell lines and in lymph nodes from a mouse model bearing a translocated *c-MYC* gene from the human BL line IARC-BL60, we surprisingly discovered a complex electrophoretic profile. Indeed, the BL60 cell line carrying the *t*(8;22) *c-MYC* translocation exhibits a simple pattern, with a single c-Myc2 isoform. Analysis of the *c-MYC* transcripts expressed by tumor lymph nodes in the mouse *λc-MYC* (*A^vy^/a*) showed for the first time five transcripts that are associated with *t*(8;22) *c-MYC* translocation. The five transcripts were correlated with the production of c-Myc2 and c-MycS, and loss of c-Myc1. The contribution of these transcripts to the oncogenic activation of the *t*(8;22) *c-MYC* is discussed.

## 1. Introduction

The human *c-MYC* proto-oncogene is involved in the control of many cellular processes including cell growth and apoptosis [1]. In normal cells, most transcripts of the human proto-oncogene *c-MYC* start at alternative promoters P1 and P2 and encode three proteins designated c-Myc1 (67 kDa), c-Myc2 (64 kDa), and c-MycS (55 kDa) [2,3,4]. These three isoforms arise by alternative initiation of translation starting at three different in-frame codons: a non-canonical CUG for c-Myc1, an AUG located 15 codons downstream for c-Myc2 [2], and an AUG 100 codons further downstream for c-MycS [3]. The resulting proteins contain the same carboxy-terminal domain but differ in their N-terminal regions. A third promoter (P3) within the first intron has also been described [5]. P3 promoter transcriptional activity leads to an mRNA that lacks the N-terminal encoding sequence (15 amino acids) that is specific to c-Myc1. Therefore, the specific function of each of the c-Myc proteins lies in their N-terminal region. The ability to express at least three amino-terminally unique forms of the c-Myc protein seems important for the normal function of c-Myc in cell growth control. Indeed, c-Myc1 and c-Myc2 possess different transactivation efficiencies at the non-canonical CCAAT/enhancer-binding protein-binding site [6]. Furthermore, c-Myc2 is predominant in growing cells while c-Myc1 is preferred as cells approach high-density growth arrest [7]. In addition, c-Myc1 exhibits a strong induction of apoptosis compared to c-Myc2. c-MycS, which is transiently expressed during rapid cell growth [3], lacks the first 100 amino acids containing two phosphorylation sites (Thr58 and Ser62) that are involved in the stability of c-Myc proteins to proteasomal degradation. Mutations of these sites are often linked to B-cell lymphomas and are correlated with reduced apoptotic potential [8]. In normal mammalian cells, these c-Myc isoforms do not accumulate singly but in specific combinations or ratios that are characteristic of a given cell status. Therefore, imbalanced expression of the different c-Myc proteins directly contributes to the loss of cell growth control that is associated with tumor development [7].

During B-cell maturation, the IG loci (IGH, IGK, and IGL) undergo sequential rearrangements orchestrated by several B-cell-specific enhancers. Their activity allows proper B-cell differentiation and secretion of high affinity antibodies. However, the activity of the enhancers can be hijacked, leading to B-cell lymphomas. Indeed, illegitimate recombination outside the IG loci can lead to the deregulation of proto-oncogenes such as *c-MYC*. Several recurrent recombinations involving the IG and *c-MYC* loci have been described in B-cell malignancies [9]. Among the mechanisms facilitating the recombination between the IG and *c-MYC* loci is their spatial proximity in human B-cell nuclei [10].

In Burkitt’s lymphoma (BL) cells that have a *c-MYC* chromosomal translocation to one of the immunoglobulin (IG) loci on chromosomes 2, 14, or 22, the transcription of a *c-MYC* proto-oncogene is characterized by preferential transcription from the *c-MYC* promoter P1 [11]. Shifted promoter P2 to P1 leads to a change in the c-Myc1/c-Myc2 ratio that is often observed in human tumors cell lines [12]. In most cases, only c-Myc2 was detected with little or no c-Myc1. Several transgenic mouse models were generated to drive *c-MYC* expression throughout B-cell development under the control of different IG enhancers, in order to mimic the *IG-c-MYC* translocation [13]. The *λc-MYC* mouse model bearing a translocated *c-MYC* gene from the human BL line IARC-BL60 (Figure 1) exhibits aggressive lymphomas with striking similarities to human BL [14]. It was suggested that the mutation within the 5′ sequence and promoter shift from P2 to P1 leads to an unbalanced expression of the different c-Myc proteins in BL60. Indeed, BL60 exhibits an increased amount of c-Myc2 and fails to synthesize the detectable levels of c-Myc1 as assessed by Western blot [2,15].

Here, we report for the first time that tumors from lymph nodes of *λc-MYC* (*A^vy^/a*) mice exhibit five transcripts as new tools to exacerbate the imbalanced expression of c-Myc proteins that may explain the aggressive lymphomas in the *λc-MYC* (*A^vy^/a*) mouse model.

## 2. Materials and Methods

### 2.1. Mouse Models

*λc-MYC* (*a/a*) mice on a C57BL/6 background were kindly provided by Pr. Georg BORNKAMM (Helmholtz Center, Munich, Germany). The *λc-MYC* (*A^vy^/a*) mouse model that was used in this work develops a human Burkitt-like lymphoma that is derived from crossing *λc-MYC* (*a/a*) mice with Yellow mice (*A^vy^/a*). *A^vy^* mice were kindly provided by Dr. David SKAAR (Department of Biological Sciences, Centre for Human Health and the Environment, North Carolina State, University, Raleigh, NC, USA) for epigenetic studies. All animal experiments and protocols were conducted in accordance with the European guidelines and regulations for animals used for scientific purposes, implemented in France as follows “Décret n°2012-118 du 1er février 2013 relatif à la protection des animaux utilisés à des fins scientifiques”. Considerable efforts were made to minimize the number of animals that were used and to ensure optimal conditions for their well-being and welfare before, during, and after each experiment.

### 2.2. Cell Lines

Burkitt’s cell lines that were used in this work (BL41 (RRID:CVCL_1087), BL2 (RRID:CVCL_1966) and P3HR1, RAJI, and Namalwa) were kindly provided by Dr Sylvie Ranger-Rogez [16], cultured from cryopreserved cells, in RPMI- 1640 medium that was supplemented with 10% fetal calf serum (FCS), 1% penicillin (100 μg/mL), 1% streptomycin (100 μg/mL), L-Glutamine, sodium pyruvate, and vitamins. All the cells were cultured in a humidified chamber at 37 °C and 5% CO_2_. They were not stored beyond the 35th passage. The cell pellets were washed with PBS (phosphate-buffered saline) twice and then used for either RNA or protein extraction.

### 2.3. RNA Extraction

RNA was prepared using an RNeasy Mini Kit (Qiagen, Les Ulis, France) according to the manufacturer’s instructions and treated with DNase I by using RNase free DNase set (Qiagen).

### 2.4. Identification of 5′UTR, 3′UTR and Full-Length of cDNA

The SMARTer RACE 5′/3′ Kit (Takara Bio, Inc. Mountain view, CA 94043 USA), Catalog no 634860) was used to perform both 5′- and 3′-rapid amplification of cDNA ends (RACE) according to the manufacturer’s instructions. Briefly, 1μg of total RNA (from lymph node tumors or cell lines) was used for 5′-RACE- Ready cDNA and 3′-RACE-Ready cDNA. Primer pairs P5′UTR1/UPM (Universal Primer A Mix) and P5′UTR2/UPM were used for PCR amplification using 5′-RACE-Ready cDNA as template to obtain 5′UTR, P3′UTR1/UPM was used to obtain 3′UTR by PCR amplification using 3′-RACE-Ready cDNA as template. By cloning and sequencing (BigDye™ Direct Cycle Sequencing Kit—Thermo Fisher Life technologies SAS Courtaboeuf Cedex/Villebon sur Yvette, France) the 5′ and 3′ UTRs, we designed specific primers to obtain full-cDNA and the corresponding mRNA structure. All the primers that were used in this work are listed in Table 1. PCR amplification was performed as follows: denaturation 2 min at 94 °C, 35 cycles (94 °C 15 s, 61 °C 15 s, 72 °C 1 min), 72 °C 2 min.

### 2.5. In Vitro Expression of c-Myc1, c-Myc2 and c-MycS Reading Frames

Reading frame sequences encoding each protein (Appendix A) were amplified by PCR (as described above) using primer pairs that are described in Table 1 and cloned in the pT7CFE1 expression vector (Appendix A). In vitro transcription and translation were performed according to manufacturer’s instruction (pT7CFE1-CHis Vector for Mammalian Cell- Free Protein Expression; Thermo Fisher).

### 2.6. Protein Extraction and Western Blot

Fresh or frozen animal tissues (30 mg) were dissected on ice. 2 × 10^6^ cells (extracted tissues or cell culture) were suspended in 50 μL RIPA (Radio ImmunoPrecipitaion Assay; BioRad) lysis buffer containing 200 mM PMSF (Alpha-Phenyl Methyl Sulfonyl Fluoride, serine protease inhibitor), 100 mM sodium orthovanadate, and a protease inhibitor cocktail (Santa-Cruz Biotech). Lysis was performed on ice for 30 min. The soluble protein fraction was recovered after centrifugation at 13,000 rpm at 4 °C for 15 min. The proteins were assayed by the Bradford method. Then, the proteins that were extracted or in vitro synthesized proteins were denatured at 95 °C for 3–5 min in the presence of β-mercaptoethanol and Laemmli blue.

Equal amounts of denatured proteins (30 μg/lane) were separated by SDS-PAGE then transferred to PVDF membrane. Nonspecific binding sites were blocked for 1 h with 5% non-fat dry milk in TBS containing 0.1% Tween-20. After overnight incubation at 4 °C with specific primary Ab (9E10, dilution 1/200, sc-40 Santa Cruz Biotechnology 92000, Nanterre, France), membranes were incubated with appropriate HRP-conjugated secondary Ab (dilution 1/1000, sc-2357 Santa Cruz Biotechnology) for 1 h at room temperature and revealed by an enhanced chemiluminescent detection method (Immubilon Western, Millipore, Merck Millipore Saint Quentin en Yvelines, France). Protein-loading control was performed with GAPDH.

## 3. Results

### 3.1. c-MYC Gene Translocation and Synthesis of c-Myc Proteins in Burkitt Lymphomas

In our *λc-MYC* (*A^vy^/a*) model mouse (Figure 1) of Burkitt lymphomas [14], we repeatedly obtained a complex electrophoretic pattern on Western blot, notably in lymph node tumors. We consistently observed two bands in those tumors that could correspond to c-Myc2 and c-MycS with an apparent molecular weight of 64 kDa and 55 kDa, respectively, in addition to a 45 kDa band (Figure 2A). The latter, in addition to c-Myc1 and c-Myc2, has been described in several human and avian cell lines [3]. All Burkitt lymphoma cell lines that were studied exhibited doublet bands (Figure 2B): c-Myc2 (64 kDa) and probably its phosphorylated [2] form (65 kDa). Burkitt lymphoma cell lines are considered to express only c-Myc2 [2]. Since we were unable to explain the presence of the other bands (Figure 2A,B), we performed a control test to detect c-Myc proteins.

### 3.2. c-Myc Protein Production by Cell-Free Translation of a Single Cloned Reading Frame

The reading frames encoding c-Myc1, c-Myc2, or c-MycS plus two other reading frames downstream (Appendix A) from the c-MycS translation initiation site (named c-MycS1 and c-MycS2) were cloned into the pT7CFE1 vector (Appendix A), and the corresponding mRNA was translated in vitro using a mammalian in vitro translation system based on HeLa cell lysates. This experimental approach yielded the proteins that were detected in Figure 2C. In each case, significant bands with apparent molecular weights of 67 kDa (c-Myc1), 64 kDa (c-Myc2); 55 kDa (c-MycS), 45 kDa (c-MycS1), and 43 kDa (c-MycS2) were obtained. Minor bands corresponding to the phosphorylated forms [2] were also detected. However, c-MycS, c-MycS1, and c-MycS2 lack (Appendix A) the first 100, 109, and 133 amino acids, respectively, which contain two phosphorylation sites. Moreover, other phosphorylation sites have been described in the C- terminal region of the c-Myc proteins [17]. This result validates the capability of the 9E10 antibody used against the C-terminal region to detect c-Myc protein isoforms with an unaltered C-terminal region, and to some extent the identity of the bands that were described in Figure 2A,B. Of note, c-Myc1 (67 kDa) was undetectable in both lymph nodes (this work) and Burkitt’s lymphoma cell lines (this work and) [2].

### 3.3. c-MYC Transcripts in Lymph Nodes from λc-MYC Mice

The *λc-MYC* mouse model was obtained by transgenesis using a 12 Kbp DNA fragment (Figure 1A) from the *c-MYC* translocation *t*(8;22) of BL60 cell line [11,14]. As the electrophoretic profile of c-Myc proteins in tumors from lymph nodes from *λc-MYC* (*A^vy^/a*) mice was no longer similar to that which was described for the BL60 line, with c-Myc2 but also c-MycS, and c-MycS1 (apparent molecular weight of 64, 55, and 45 kDa respectively, Figure 2A), we then analyzed the *c-MYC* transcripts in these tumors.

Total RNA was extracted and the 5′ and 3′ regions were obtained by RACE-PCR. The *c-MYC*-specific primers that were used are described in Table 1. To obtain the 5′UTR (5′UnTranslated Region), we used the primers P5′UTR1 and P5′UTR2 located at exon 1 and exon 2, respectively (Figure 3A), to account for the different promoter activities (P1, P2, or P3) that were described in the literature. The primer P3′UTR1 (Figure 3A) was used to obtain 3′UTR. The 5′ and 3′ established regions allowed us to define PFull1, PFull2, and PFull3 primers (Figure 3B,C) to obtain the total cDNA and the structure of *c-MYC* mRNAs after sequencing. For the 3’ region, we repeatedly obtained (Appendix A) the same 3’UTRs for both the lymph nodes and for the BL41 cell line corresponding to the use of the two described polyadenylation sites [18].

Sequence analysis of the cDNA revealed the presence of five *c-MYC* transcripts (Figure 3), two from transcription through P1 (P1 promoter) and three from P3 (P3 promoter) which is located at intron 1 (Figure 3A). Transcription from P1 has been reported in the BL60 line, but to our knowledge, the nature of the transcripts has not been described. Transcription from P3 has not been reported in the BL60 cells. It is noteworthy that Trans1 (transcript 1) carries the PvuII mutation which has been suggested to be responsible for the production of the single c-Myc2 variant in BL60 cells (Figure 3 and Appendix A). All *c-MYC* transcripts that were described in this work differ at the 5’UTR and/or at exon 2 (Figure 3). They all have the entire last exon (exon3). In addition, the 5′UTRs of three transcripts (Trans1,2,3) contains termination codons in all three reading frames whereas the two 5′UTR transcripts (Trans4,5) (Appendix A) are in reading frames of a part of exon 2 and the coding region of c-MycS.

### 3.4. P1 Promoter Transcripts and c-Myc Proteins

The genomic region corresponding to transcripts 1 and 2 (Trans1 and Trans2) resulting from P1 promoter use (Figure 3 and Appendix A) revealed that the mature Trans1 mRNA (Figure 3 and Appendix A) is derived from the canonical splicing of a *c-MYC* pre-mRNA (Figure 3B and Appendix A) that was previously described [15]. However, Trans2 (revealed in this work) is obtained by an alternative splicing acceptor site within exon 2 (Figure 2B and Appendix A). In both cases, the donor (GT) and acceptor (AG) sites are canonicals. Translation of the two transcripts in the three reading frames predicts (Figure 3) the formation of c-Myc1 (Trans1) and c-MycS (Trans2) isoforms. Nevertheless, our Western blot experiments (Figure 2A) do not detect the c-Myc1 (p67) isoform. This result is in agreement with previously reported data showing that a substitution near the non-canonical CUG initiation site prevents translation from the CUG site and thus c-Myc1 production [2]. It is, therefore, very likely that the translation of Trans1 leads to the formation of c-Myc2 from the AUG initiation codon that is located at exon 2 (Figure 3 and Appendix A). Taken together, analysis of the two transcripts coupled with Western blot data (Figure 2A) suggest production of c-Myc2 (instead of c-Myc1) and c-MycS by in vivo translation of Trans1 and Trans2, respectively. The detection of a smaller isoform (45 kDa: c-MycS1) could originate from the translation of Trans1 and/or Trans2 transcripts. Two arguments: the leaky scan of the c-Myc transcript described previously [2], the detection of an isoform of the same size in several human and avian lines and the actual in vitro translation (Figure 2C) support this suggestion. The last open reading frame (Appendix A) of the c-Myc transcript (c-MycS2,), although it is translatable (Figure 2C) in vitro (43 kDa), does not seem to translate in vivo. Finally, without excluding hyperphosphorylated isoforms of the c-Myc proteins, we do not have a reasonable explanation to interpret the high molecular weight bands at present.

### 3.5. P3 Promoter Transcripts and c-Myc Proteins

A similar analysis of the P3 promoter transcripts (Trans3, Trans4 and Trans5) revealed (Figure 3C and Appendix A) that the Trans4 transcript is derived from the splicing of a pre-mRNA with canonical donor and acceptor sites that are located at intron 1 while respecting the canonical sites between exon 2 and exon 3. Since Trans4 lacks exon 1 (Figure 3C), translation of this mRNA in vivo will necessarily lead to the production of c-Myc2. The two other transcripts (Trans3 and Trans4) were derived from alternative splicing of pre-mRNA using different non-canonical splice donor and acceptor sites at intron 1 and exon 2. If these mRNA were translated in vivo, they would produce at least the c-MycS isoform. Overall, P3 transcripts will result in the enrichment of c-Myc2, c-MycS, and for the same reason as suggested above, c-MycS1 isoforms.

### 3.6. c-MYC Transcripts and c-Myc Proteins in the BL41 Cell Line

Data that were described above led us to carry out a study of *c-MYC* transcripts and the corresponding proteins in a Burkitt lymphoma cell line.

Total RNA were extracted and subjected to the same RACE-PCR approach to obtain the 5′UTR region and then the total *c-MYC* mRNA expressed by BL41 cells. We used the same primers as above to obtain the 5′ and 3′ UTRs. We obtained amplifications with primer P5′UTR2 located in exon 2 (for 5′UTR) and primer P3′UTR1 located in exon 3 (for 3′UTR). We did not obtain any amplification with the other primer P5′UTR1 located in exon 1.

Sequence analysis of cDNA corresponding to the mRNA showed a single repeatedly obtained transcript (Figure 4 and Appendix A). The 5′UTR was formed by a hybrid sequence including an IGH region and the end of intron 1 of *c-MYC*. The rest of the mRNA was transcribed from exons 2 and 3. This hybrid transcript was probably the result of transcriptional activity at the IGH locus following the *t*(8;14) translocation that characterizes the BL41 cell line. The fact that we did not obtain any amplification with the P5′UTR1 primer supports the presence of only *c-MYC* transcripts lacking exon 1. The translation of this mRNA predicts c-Myc2 protein in agreement with previously described data and with the results that were obtained by Western blot (Figure 2B).

## 4. Discussion

In this study, we report for the first time that tumor lymph nodes from the *λc-MYC* (*A^vy^*/*a*) mice exhibit five different *c-MYC* mRNA that invariably lead to the production of c- Myc2 and/or c-MycS previously described and likely a new isoform c-MycS1 shown on Western blot.

It is now well established that the quantitative and qualitative expression of the two proteins c-Myc1 and c-Myc2 and to a lesser extent c-MycS play an important role in the control of normal (non-cancerous) cell proliferation and cell homeostasis [19]. Although the ratio between these three isoforms is difficult to determine because their quantity changes according to stages of the cell cycle, the mechanisms that support the production of c-Myc2 and/or c-MycS promote a cancerous state: The *c-MYC* proto- oncogene becomes the *c-MYC* oncogene [2,19].

Virtually all Burkitt lymphoma cell lines show a rearrangement between the *c-MYC* locus and one of the immunoglobulin gene loci. These rearrangements alone or in combination with mutations in the 5′ region of the *c-MYC* gene often lead to the production of c-Myc2 alone. Previously, it was shown that the BL60 line carrying the *t*(8;22) translocation exhibits a shift from promoter 2 to promoter 1 and harbors a substitution that prevents translation from the upstream non-canonical CUG translation site. Thus, BL60 was described to have lost c-Myc1 production and to produce only c-Myc2 [2]. It should be noted that the 12 kbp DNA fragment (Figure 1A) from the *t*(8;22) translocation of the BL60 line that was used to obtain the *λc-MYC* mouse [14] contains all the genetic information (including the *c-MYC* gene) that is necessary for the development of an aggressive B lymphoma, particularly in the lymph nodes. In agreement with these previous data, the Trans1 mRNA that was discovered in this work was derived from the P1 promoter (shift from P2 to P1, as in BL60) and carries the described substitution near the CUG translation site as reported for translation of the only open reading frame of Trans1 which codes for the production c-Myc1. Nevertheless, its presence in Western blots remains undetectable. This observation supports the fact that a mutation near the CUG translation site prevented its use as a translation initiation site and strongly suggests that in vivo translation of Trans1 leads to the formation of c-Myc2.

The second mRNA (Trans2) that we found is also from the P1 promoter, also carries the substitution near CUG, but with alternative splicing at exon 2. The only c-Myc protein that will be produced if this Trans2 is translated in vivo will be c-MycS. This hypothesis is in agreement with our Western blot data.

The other three transcripts (Trans3, 4, and 5) were derived from transcription at the P3 promoter that is located in intron 1. The translation of Trans3 predicts the production of c-Myc2 according to the only open reading frame. However, the predictions that can be drawn from the primary structure of Trans4 and 5 are less clear. Indeed, despite different alternative splicing, Trans4 and 5 each present an open reading frame from, respectively, the first and second nucleotides of transcription initiation to the stop codon at the last exon 3 of the *c-MYC* gene. Several potential non-AUG [20] sites were located in the 5′ region of these two mRNA but their use as translation initiation sites remains to be determined experimentally. Finally, we did not identify a specific transcript that could produce the new c-MycS1 isoform suggested in Western blot. Nevertheless, all the transcripts that were described in this work are likely to produce this isoform by the leaky scanning mechanism that has been proposed for the translation of the *c-MYC* transcript [2].

Alternative splicing of pre-mRNA and/or alternative use of promoters play an important role in the genesis and development of different types of cancer. Indeed, many cases of splicing isoforms that lead to or promote cancer progression have been identified. For example, cyclin D1b, the oncogenic form of cyclin D1, expressed in lymphomas, results from alternative splicing. Another example is the anti-apoptotic isoform Bcl-xL which is strongly expressed in a large number of tumors to ensure high cell survival [21]. Finally, the use of alternative promoters and/or alternative splicing generates a complex pattern of pro- and anti-apoptotic isoforms of the TP73 gene [22] which is strikingly similar to the mechanism that is used by the *t*(8;22) translocation as found in this study, to generate only oncogenic forms of c-Myc.

In summary, we report for the first time the concept of multi-transcript genesis as a tool for the development of Burkitt’s lymphoma in the case of *t*(8;22) translocation. Our work suggests that this type of approach may be useful for the identification of *c-MYC* transcripts with B lymphoma. Of course, these transcripts will differ depending on the type of B lymphoma that is associated with specific c-MYC dysregulation. The BL41 line that was used in this work is an example.

In this context, an exhaustive study of *c-MYC* transcripts in Burkitt lymphoma patients as well as in other animal models becomes necessary. A nanopore sequencing approach is currently underway and the data obtained will provide new diagnostic tools. We posit that this comprehensive characterization of c-Myc transcripts will contribute to reduced expression of oncogenic c-Myc forms in a targeted manner. Indeed, today it is quite possible to target in vivo translation of a specifically characterized transcript using antisense oligonucleotides as therapeutic agents of c-Myc-addicted tumors [23].

## Figures and Tables

**Figure 1 biomolecules-12-00836-f001:**
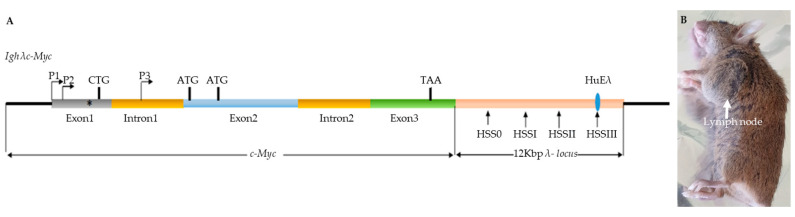
(**A**) Structure of the *λc-MYC* transgene; (**B**) carried by the *λc-MYC* (*A^vy^/a*) model mouse of Burkitt’s lymphoma. Horizontal arrows indicate the two *c-MYC* promoters, P1 and P2, located in the first noncoding exon. Vertical arrows indicate the locations of the four previously defined [14] DNaseI-hypersensitive sites (HSS) in the IglL; CTG, ATG, and ATG shown represent the previously described translation initiation sites for c-Myc1, c-Myc2, and c-MyS, respectively. Diagram is not to scale. *: PvuII mutation.

**Figure 2 biomolecules-12-00836-f002:**
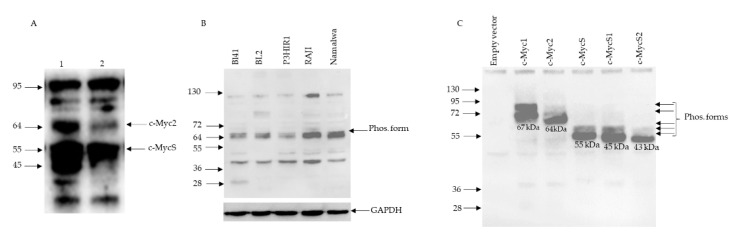
Analysis of the migration profile of c-Myc proteins by Western blot using the monoclonal antibody 9E10 directed against the conserved C-terminal part of c-Myc1, c-Myc2,c-MycS, and two other potential reading frames that were not previously described and named here c-MycS1 and c-MycS2 (Appendix A). (**A**) Typical results were obtained from cell extracts that were prepared from lymph nodes from two *λc-MYC* (*A^vy^/a*) mice (lane 1, lane 2). (**B**) Results that were obtained from extracts that were prepared from BL cell lines. (**C**) Electrophoretic pattern of c- Myc1, c-Myc2, c-MycS, c-MycS1, and c-MycS2 proteins, obtained in vitro by transcription and translation of the open reading frames (Orf-c-Myc1, Orf-c- Myc2, Orf-c-MycS, Orf-c-MycS1, and Orf-c-MycS2) cloned separately into the pT7CFE1 vector (Appendix A). The apparent molecular weight of the different bands is shown on the left; Empty vector: Negative control; Phos. Form: Phosphorylated form; GAPDH: loading control.

**Figure 3 biomolecules-12-00836-f003:**
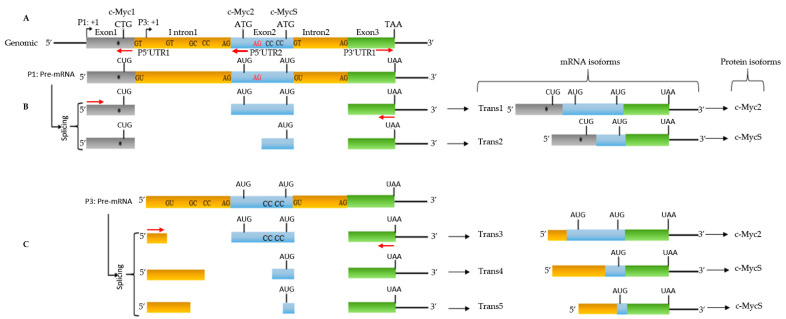
Different events leading to the genesis of several mRNAs of the *c-MYC* gene related to genomic structure. Genomic structure (**A**) at the origin of the different events (**B**,**C**). Primers P5′UTR1 and P5′UTR2 that were used to obtain the 5′UTR; primer P3′UTR1 to obtain the 3′UTR, and the other primers were used as indicated to obtain the full-length cDNA. The dinucleotides indicated represent the different splice donor and acceptor sites. The triplets indicate the translation initiation codons to obtain c-Myc1, c-Myc2, or c-MycS. TAA: Translation stop codon. *: PvuII mutation. Diagram is not to scale.

**Figure 4 biomolecules-12-00836-f004:**
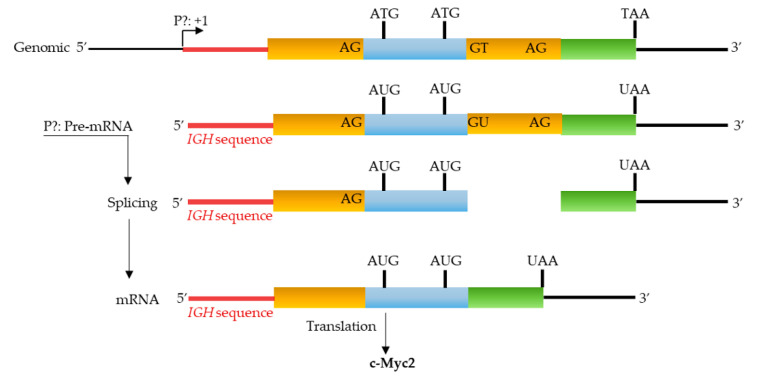
Partial genomic structure of the c-MYC gene that is associated with *t*(8;14) translocation of BL41. The expression from a yet to be determined promoter (P?) generates a hybrid mRNA with an IGH sequence. The translation of the resulting mRNA leads to the production of the c-Myc2 protein. Diagram is not to scale.

**Table 1 biomolecules-12-00836-t001:** Primers (P) that were used to obtain 5′UTR, 3′UTR and total length cDNAs of *c-MYC* transcripts or only open reading frames (ORFs) are indicated. The underlined sequences CATATG of Orf-c-MycS, present in all of-c-Myc, corresponds to NdeI restriction sites; the underlined sequence of Orfs corresponds to XhoI restriction site; CTG and ATG: Translation start codons; TTA: reverse complement Stop Codon.

Primer (P)	5′ to 3′ Sequence	Localisation	Use For
P5′UTR1	CTGGTTTTCCACTACCCGAAA	Exon 1	5′UTR Amplification
P5′UTR2	CGTTGAGGGGCATCGTCGCGGG	Exon 2	5′UTR Amplification
P3′UTR1	CTACGGAACTCTTGTGCGTAA	Start at the End of the Last Exon	3′UTR Amplification
PFull1	GACCCCCGAGCTGTGCTGCTC	Start at P1:+1	Full-Length cDNA
PFull2	GGGAACAGCCGCAGCGGAGGG	Start at P3:+1	Full-Length cDNA
PFull3	TTACGCACAAGAGTTCCGTAG	Start at the End of the Last Exon	Full-Length cDNA
PFull4	GGAGATAGTGGGGCTCAGAGC	Start at P?:+1	Full-Length cDNA
Orf-c-Myc1	ATCATATGCTGGATTTTTTTCGGG	CTG Initiation Codon	Full-Length Orf-c-Myc1
Orf-c-Myc2	ATCATATGCCCCTCAACGTTAGCT	ATG Initiation Codon	Full-Length Orf-c-Myc2
Orf-c-MycS	ATCATATGGTGACCGAGCTGCTG	ATG Initiation Codon	Full-Length Orf-c-MycS
Orfs	ATCTCGAGTTACGCACAAGAGTTC	TTA Stop Codon	Reverse for all Orfs

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
