# Peer review of "Alternative c-MYC mRNA Transcripts as an Additional Tool for c-Myc2 and c-MycS Production in BL60 Tumors"

_biomolecules, 2022, doi:10.3390/biom12060836_

Round 1
Reviewer 1 Report
- Please describe more clearly in the discussion the possible relevance of the results in the mouse model for clinical aspects.
- "In this context, an exhaustive study of c-MYC transcripts becomes necessary." Please describe the character of such investigations more clearly, another animal model study? a study in Burkitt lymphoma patients?
- For the Discussion: Would you consider multi-transcript genesis a frequent phenomenon in the case of BL? Or do you consider this phenomenon to be a single case observation? Are there any results from the literature that could facilitate the interpretation of this phenomenon? Maybe deriving from other lymphoma subtypes with reciprocal rearrangements?
- Considering the broader readership of the journal, the authors may consider shortening their manuscript to a shorter format (e.g. a communication or brief report), which would require the transfer of some sections to the online supplement.
Author Response
Please see the attachement

Reviewer 2 Report
In this manuscript, Ibrahim et al, perform a detail study on MYC protein describing the transcriptomic structure and how this is translated to the specific isoforms. MYC is a relevant oncogene in cancer, it is overexpressed via enhancer hijacking in virtually all Burkitt lymphoma patients. Therefore this study expand the acknowledge of MYC protein and is relevant in the lymphoma field.
Mayor review
- In BL, the MYC transcript originate from the translocated allele and is associated with the location of the translocation breakpoints. This could explain in a subset of the cases that the main isoform observed was MYC 2 at least in the BL cases with IGH::MYC translocation that have the frequently the breakpoints located in exon 1 or intron 1. The authors should discuss in details this relevant points that are recently described. Moreover in the context of BL 60 harboring IGK::MYC that typically display breakpoints downstream of MYC.
- Identify of the cell lines is missing in the material and method section. The authors must perform identity text for cell lines used in the manuscript, this is mandatory when you are working with cell lines.
-Looking in detail exon 2 of MYC there are a second ATG start downstream of the MYC-S start codon protein is important to differentiated from the other? What is the significant value of this? The authors should address this issue because could explain the additional WB bands observed in Figure 2.
-The Western Blot experiments showed a band around 45 KDa? Could the authors mention hypthesis why? Something related with the previous comment? Furthermore the authors could investigate more in detail the protein isoform using mass spectometry experiments to resolve the unexpected bands. Even an essential control experiment in a cell line lacking MYC protein to text the specify of the antibody is missing. The last experiment is crucial to rule out unspecific bands.
- Regarding the Western Blot, in all the figures the loading control (GADH, based on the method section) is missing. The authors should include this in all the figures displaying WB results.
- Results section 3.2 the authors mentioned that the cloning the three different MYC isoform into the pT7 CFE1 vector confirmed their hypothesis about the WB bands observed in WB experiment performed using protein extraction from lymph nodes and BL cell lines. However the authors should clarify which is the hypothesis and what their refer. Besides, the authors should mention why still in the empty vector there are some weak bands.
- Authors used a the λc-MYC mouse model obtained by transgenesis using a 12 Kbp DNA fragment from the c- MYC translocation t(8;22) of BL60 cell line. Therefore would be important to investigate the MYC protein by WB in the BL60 cell line, to see which isoform could be detected.
- In line with the first comment, the authors should investigate if the BL41 has the translocated breakpoint in intron 1 and this could explain why exon 1 is not detected. Because in the experiments the authors detected the der(14). On the other hand there are no expression of untranslocated MYC?
Minor review
- Figure legend 1 is not completed.
- To follow the manuscript is better to refer figure 1 in advance in the introduction to improve the understanding of the manuscript.
- Result section in chapter P1 promoter transcripts and c-Myc proteins, better used 67KDa instead to p67, that was not used previously.
Round 2
Reviewer 1 Report
the revised version of the manuscript seems to be acceptable for publication to my point of view. The authors considered all relevant critizisms of the reviewers.Thanks.